# Comprehensive Transcriptome Analysis of Different Skin Colors to Evaluate Genes Related to the Production of Pigment in Celestial Goldfish

**DOI:** 10.3390/biology12010007

**Published:** 2022-12-21

**Authors:** Rongni Li, Yansheng Sun, Ran Cui, Xin Zhang

**Affiliations:** Fisheries Science Institute, Beijing Academy of Agriculture and Forestry Sciences, Beijing 100097, China

**Keywords:** celestial goldfish, skin color, transcriptome, pigment production

## Abstract

**Simple Summary:**

Skin color is the result of various pigments synthesized by dyeing groups, while goldfish have a greater variety of pigment cells and a wider range of skin colors. Up to now, several pigment cells have been reported, including melanophores, erythrocytes, xantho-phores, iridophores, leucophores and cyanophores. Therefore, the key genes associated with sieve pigments will provide clues for the production of goldfish with a diversity of skin tones, benefiting the goldfish industry. In this study, high-throughput sequencing was performed on the back skin tissue of celestial goldfish with different skin colors (white, yellow, brown), and it was found that melanin production, tyrosine metabolism pathway, Wnt signaling pathway, and MAPK signaling pathway may play an important role in pigmentation. The study provides valuable data for screening candidate genes involved in pigment formation, and the results of this study will lay a foundation for further study on the expression characteristics and gene network analysis of pigment related genes.

**Abstract:**

Skin color is an important phenotypic feature of vertebrate fitness under natural conditions. Celestial goldfish, a common goldfish breed in China, mainly shows three kinds of skin colors including white, yellow and brown. However, the molecular genetic basis of this phenotype is still unclear. In this study, high-throughput sequencing was carried out on the back skin tissues of celestial goldfish with different skin colors. About 58.46 Gb of original data were generated, filtered and blasted, and 74,297 mRNAs were obtained according to the reference transcriptome. A total of 4653 differentially expressed genes were screened out among the brown, yellow and white groups, and the expression of melanogenesis related genes in brown goldfish was significantly higher than the other two groups. There are 19 common differentially expressed genes among three groups, of which eight genes are related to pigment production, including *tyrp1a*, *slc2a11b*, *mlana*, *gch2*, *loc113060382*, *loc113079820*, *loc113068772* and *loc113059134*. RT-qPCR verified that the expression patterns of randomly selected differentially expressed transcripts were highly consistent with those obtained by RNA sequencing. GO and KEGG annotation revealed that these differentially expressed genes were mostly enriched in pathways of the production of pigment, including melanogenesis, tyrosine metabolism, *Wnt* signaling pathway, *MAPK* signaling pathway etc. These results indicated that the external characteristics of goldfish are consistent with the analysis results at transcriptome level. The results of this study will lay a foundation for further study on the expression characteristics and gene network analysis of pigment related genes.

## 1. Introduction

Skin color is the result of diverse pigments synthesized by chromatophores; different kinds of pigment cells are found in various vertebrates, while there are more kinds of pigment cells and a wider range of skin colors in goldfish. Up to now, several pigment cells have been reported, including melanophores, erythrocytes, xanthophores, iridophores, leucophores and cyanophores [1,2]. Although the production of skin color in fish is complex and varies depending on the breed, feed, and water environment, genes are the most basic and essential element that influences skin pigmentation.

RNA-seq is a study of gene structure and function at the overall transcriptome level, based on high-throughput sequencing technology, and reveals the molecular mechanism in specific biological processes [3]. Many studies have been devoted to explaining the genetic basis of pigment production by RNA-seq in recent years, and a certain number of important genes and biological pathways have been found, including melanogenesis and tyrosine metabolism. Jiang et al. [4] performed transcriptome analysis of skin color during and after the overwintering of red tilapia, showing that pigmentation-related genes include *tyrp1*, *tyr*, *pmel*, *MITF* and apoptosis and autophagy-related genes possibly regulate the molecular mechanisms of color change. Gan et al. [5] performed comparative transcriptome analysis on four skin colors of goldfish and concluded that the melanogenesis and tyrosine metabolism pathway may affect the skin pigment agglutination process. Luo et al. [6] found that biological pathway was mainly enriched in tyrosine metabolism and melanogenesis through transcriptome analysis of koi fish with white, brown and red skin colors. Chen et al. [7] reported that, consistent with goldfish, the differential expressed genes of various skin colors in Silurus meridionalis is also mainly concentrated in melanogenesis, melanin metabolism and melanin agglutination hormone activity pathway. In addition, *tyr*osinase (*TYR*) [8,9,10], solute carrier family 24 member 5 [11,12], GTP_cyclohydroI (*gch*) [13,14] and microphthalmia associated transcription factor (*MITF*) [15] have been reported as important genes involved in melanogenesis. Celestial goldfish, as a unique variety in China, is deeply loved by people with eyes turned upward and colorful body colors. However, the key genes and biological processes, which are involved in the production of pigment in celestial goldfish, are still unclear.

The molecular process of the production of pigment in celestial goldfish has not yet undergone a comprehensive examination. Therefore, clarifying the roles of the differential expression genes can contribute to understanding the changes associated with the production of pigment in different skin color groups. The purpose of this study is to obtain an overview of transcriptome profiles in three kinds of skin colors including white, yellow and brown in skin tissue. In addition, we identified differentially expressed genes that were possibly involved in the production of pigment and screened out common differential expression genes among three groups. In-depth knowledge of the molecular processes underlying the production of pigment in celestial goldfish is provided in this report.

## 2. Material and Method

### 2.1. Animal Sample Collection

The celestial goldfish used in the experiment was taken from Xiaotangshan Aquaculture Base, Fisheries Research Institute, Beijing Academy of Agricultural and Forestry Sciences (Beijing, China). The experimental group used three brown and three yellow goldfish, and the control group used three white goldfish. All three groups of goldfish came from half-sibs, were raised under the same conditions until they were 4 months old, and then were slaughtered on the same day. The fish body length is 5–6 cm, and the goldfish samples collected in the observation experiment were faded and the body color was stabilized. The skin tissues from the backs of nine goldfish were extracted and stored in liquid nitrogen or at −80 °C for later use.

### 2.2. RNA Extraction and Sequencing

According to the instructions, skin tissue samples in three groups were taken, and the total RNA of each sample was extracted by RNA extraction kit (Invitrogen, Carlsbad, CA, USA). To analyze the integrity of the extracted total RNA and whether there was DNA contamination, 10 g/L agarose gel was used. The purity of RNA was detected by nucleic acid protein determination spectrophotometer (IMPLEN, Munich, Bavaria, Germany). The Agilent 2100 bioanalyzer was used to accurately detect the integrity and total amount of RNA. According to the instructions, NEB-Next Ultra TM RNA Library Prep Kit (Illumina) was used to build the library. The cDNA was synthesized by magnetic bead enrichment method, and the cDNA end was repaired, amplified and purified to obtain the cDNA library of the skin tissue of celestial goldfish.

### 2.3. Bioinformatic Analysis

The original data were filtered by removing N-containing, jointed and low-quality reads, and the GC content, Q20 and Q30 of clean data were calculated. *Carassius auratus* reference genome and gene model annotation files were downloaded from the NCBI website (https://www.ncbi.nlm.nih.gov/assembly/GCA_003368295.1 (accessed on 1 January 2022), and the parametric analysis of transcriptome sequencing data was performed. HISAT2 [16] was used to align paired-end clean reads to the reference genome. The mapped readings of each skin sample were put together using StringTie [16] with a reference-based method, and then new transcripts were annotated by Pfam [17], GO [18], KEGG [19] and other databases, and the new genes were predicted. Using the Feature-Counts program, the readings mapped to each gene were calculated [20], and the FPKM of each gene (the expected number of every thousand base segments of transcript sequence segments sequenced per million base pairs) was calculated based on the gene length. Differential expression analysis was performed using DESeq [21], with a default significant difference of two fold changes and a *q* value of 0.05.

### 2.4. Gene Ontology and Kyoto Encyclopedia of Genes and Genomes Enrichment Analysis

Cluster-Profile software was used to analyze the enrichment of differentially expressed genes by GO function and KEGG pathway, and the significant enrichment threshold was *p* value < 0.05. The main biochemical metabolic pathways and signal transduction pathways involved in color change were screened out.

### 2.5. Validation of Differentially Expressed Transcripts

RT-qPCR verified the expression pattern of 10 differentially expressed transcripts randomly selected from the skin tissue of the back. Total RNA of the sample was reverse transcribed into cDNA by prime script TM 1st stand cDNA synthesis kit. *β-actin* was selected as endogenous control gene (Appendix A). Three biological replicates and triple reactions for each sample were used for all qPCR validations. The relative expression of each gene was calculated by 2-△△Ct method, and the log2 value of multiple expression between the two groups was calculated. The column chart was drawn together with the RNA-seq sequencing results, and the trend was compared.

### 2.6. Analytical Statistics

The means and standard error of the means (SEM) are used to express data. In order to control the error rate, Hochberg and Benjamini methods are used to adjust the *p* value; differences were deemed statistically significant at *p* < 0.05.

## 3. Results

### 3.1. Blast Analysis of Transcriptome Sequencing

The transcriptome profiling of the skin tissue from goldfish of different colors were obtained to evaluate the genes involved in the production of melanin. A total of 58.46 Gb of filtered base was obtained, and the base amount of each sample reached more than 5.96 Gb, and 87.72~92.96% of these readings were specifically compared with the reference genome (https://www.ncbi.nlm.nih.gov/assembly/GCA_003368295.1, accessed on 1 January 2022). The base error rate of each sample sequencing is less than 0. 03%, and Q30 is greater than 93.90%, which indicates that the sequencing data quality is reliable (Appendix A).

Celestial goldfish is a non-model species, and its genetic annotation in production is usually not perfect. After the alignment was completed, the annotated transcription regions of the genome were searched, and the aligned sequences were assembled, so as to mine new transcripts or new genes in the species. In this study, new transcripts were assembled and annotated in Pfam, GO and KEGG databases and 2205 new genes were predicted and annotated (Appendix A).

### 3.2. Differentially Expressed Genes in Goldfish with Different Skin Colors

Through the comparison of different skin color groups, the differentially expressed genes are shown (Figure 1a,b). When compared between brown and yellow groups, a total of 1000 differentially expressed genes were screened out, of which 651 were up-regulated and 349 were down-regulated. When comparing between white and brown groups, we screened out 3033 differentially expressed genes, of which 1196 genes were up-regulated and 1837 genes were down-regulated. Consistent with the visual contrast between the brown and white groups, the number of differentially expressed genes in this group is more than other groups through RNA-seq. When comparing between white and yellow groups, a total of 1942 differentially expressed genes were screened out, of which 761 were up-regulated and 1181 were down-regulated. Some DE mRNAs were specifically expressed in the brown group, including: *loc113077208*, *loc113119661*, *loc113085971*, *loc113066815*; Some DE mRNA was specifically expressed in the white group, including: *clic5a*, *loc113104870*, *loc113118795*; There are also some specific expression of DE mRNAs in the yellow group, including: *slc2a11b*, *asz1*, *loc113092873*. These genes may regulate the production of pigment in goldfish (Appendix A).

The differentially expressed genes of three groups of goldfish were clustered and drawn into a heat map (Figure 1c). It can be seen that the expression patterns of the brown group and the yellow group are more similar, while the white group and the other two groups are at a far distance.

### 3.3. Functional Analysis of Differentially Expressed Genes

GO enrichment analysis was conducted between brown, yellow and white group according to three categories: biological process, cell components and molecular functions. The most significant 30 entries were selected from the GO enrichment analysis results to draw scatter plots (Figure 2, Appendix A). GO enrichment entries in the white and yellow groups are significant in the biological process (BP), including peptide cross-linking (GO:0018149), small molecule biosynthetic process (GO:0044283), protein tyrosine phosphatase activity (GO:0004725), etc. The GO entries of the brown and yellow groups, white and brown groups were both significantly enriched in the cellular component (CC), mainly including melanosome transport-related: troponin complex (GO:0005861), myofilament (GO:0036379), contractile fiber (GO:0043292), etc. GO enrichment concluded that differentially expressed genes were mainly enriched in the pathways related to *tyr*sine metabolism and melanosome transport.

The differentially expressed genes of three groups in goldfish were compared to KEGG database for pathway enrichment analysis, and the most significant 20 KEGG pathways were selected to draw columns (Figure 3, Appendix A). According to the results of KEGG enrichment, KEGG pathways among three groups are all enriched to pigment production related, including *Wnt* signaling pathway (dre04310), tyrosine metabolism (dre00350), melanogenesis (dre04916) and *MAPK* signaling pathway (dre04010). These results reflect that the difference of skin color of goldfish may be the result of the regulation of multiple signal pathways, and the different skin color characteristics in this study are closely related to pigment production.

### 3.4. Common Expression Gene Analysis

According to the overlapping of differential genes among different skin color groups, the results showed that 19 differential genes belonged to the common differential genes among three groups (Figure 1d, Table 1 and Appendix A), among which eight genes were reported to be related to pigment production and regulation [8,14,22,23,24,25,26], including: *tyrp1a*, *slc2a11b*, *mlana*, *gch2*, *loc113060382*, *loc113079820*, *loc113068772* and *loc113059134*. The remaining genes mainly regulate cell proliferation, differentiation and apoptosis, including transcription elongation factor a protein 3, translation initiation factor if-2, RAS and EF-hand domain containing, translocation protein *Sec62*, et al.

### 3.5. Validation of Differentially Expressed Genes Expression

Eight genes were randomly selected from the differentially expressed genes for qPCR verification, and the results showed that the gene expression trend obtained by qPCR was basically consistent with the results of RNA-seq (Figure 4). It indicates that the sequencing results of transcriptome in this study are reliable, and the differentially expressed genes among different groups can be accurately detected.

## 4. Discussion

### 4.1. Composition Analysis of Pigment Cells in Different Skin Colors

Skin color plays an important role in animal courtship, escape from enemies and signal communication, and is one of the most diverse forms of kinvertebrates. Due to diverse color pattern, goldfish could be a perfect model to study the genetic mechanism of pigmentation. The research on the formation mechanism of fish skin color provides an important theoretical reference for the study of physiology, pathology and genetics of human and other biological pigment cells. Although the production of skin color in fish is complex and varies depending on the breed, feed, and water environment [27,28], genes are the most basic and essential element that influences skin pigmentation. In this study, GO enrichment reflected the different components of pigment cells in different skin colors. The skin cells of white goldfish are mainly composed of white pigment cell, while those of yellow goldfish are mainly yellow pigment cells. Studies have reported that that characteristics and differentiation processes of white pigment and yellow pigment cells are similar in medaka [23], which also explains why there is less difference in cellular composition between white and yellow groups than in comparison with the brown group. Inconsistent with white and yellow groups, there were dendritic melanocytes in the skin of brown groups, and GO enrichment enriched the related pathways with protein tyrosine phosphatase activity. Tyrosinase (*TYR*) is the rate limiting enzyme for melanin synthesis, and its gene mutation will cause albinism in animals. Since the cloning of human tyrosinase gene in 1987 [29], much research has been carried out on the relationship between tyrosine gene and skin color (mainly albinism) in humans and animals [30,31]. Melanin is a polymer synthesized by tyrosine hydroxylation by tyrosinase catalyzed in vivo, which involves three enzymes in the tyrosinase gene family, tyrosinase (*TYR*), tyrosinase-related protein 1 (*TRP1*) and tyrosinase-related protein-2 (*TRP2*). After melanin is produced, it travels from the dendritic tip of melanocytes to keratinocytes. Furthermore, GO enrichment also enriched the related pathways with the structure of iridescent cells, actin and enzyme activity. It has been reported that the directional movement of pigment particles is closely related to actin.

### 4.2. Regulated Genes in Brown Skin Indicated the Molecular Mechanism of the Melanogenesis Pathway

At present, most studies on the skin color of goldfish are focused on pigment cell metabolism, pigment deposition and production, especially melanin production and transport, which is consistent with the results in this study. For example, Jiang et al. [32] performed comparative transcriptome analysis on the skin colors of Xingguo Red Carp and Yellow River Carp and also concluded that melanogenesis pathway affected the skin pigment agglutination process in common carp. According to our research, the differentially expressed genes were not only enriched in the melanogenesis pathways mentioned above in the existing studies, but *MAPK* signaling pathway and *Wnt* signaling pathway were also enriched. *Wnt* signaling pathways, particularly *Wnt1*, *Wnt3a* and *Wnt5a*, have been reported to play an important role in the development of melanocytes [33,34,35,36]. *Wnt3a* has been reported to reduce the number of neuronal cells and glial cells in quail, while significantly increasing the number of melanocytes [34]. Another report on zebrafish suggests that after injection of domain-inactivated *Wnt1*, the *Wnt* signaling pathway is blocked, the number of pigment cells is reduced, and neural crest cells are easily differentiated into neuronal cells and glial cells [35]. Vertebrate melanocytes almost always originate from the neural crest, and the cells derived from the neural crest undergo obvious differentiation until they are dispersed, and produce peripheral neurons, glial cells and melanocytes at precise locations [36]. Furthermore, phosphorylation of *ERK* in *MAPK* signaling pathway inhibits the production of melanin in zebrafish, while inhibition of phosphorylation of *ERK* can promote the production of melanin in zebrafish by increasing the expression of *MITF* genes, thereby upregulating downstream genes *TYR*, *TRP1* and *TRP2* [37]. The expression of *MITF* was also able to differentiate pluripotent stem cells into melanocytes in medaka [38]. Therefore, *MAPK* signaling pathway is involved in the activation of melanocyte receptors and the complex mechanism of ligand activation up-regulates *MITF* by binding to the extracellular domain of the receptor [38,39], which can effectively regulate the development of melanocytes.

### 4.3. Identification of Candidate Genes Related to the Production of Pigment

According to the analysis of sequencing data, there are 19 common differentially expressed genes in three groups, most of which are related to pigment production. The expression levels of 12 genes were the highest in the yellow group, but the lowest in the white group, most of which have the function of promoting the aggregation and diffusion of yellow pigment. For example, *loc113059134* involves perilipoprotein, which is reported to promote pigment aggregation and is highly expressed in yellow pigment cells that mediate the aggregation and transproduction of red/yellow pigments [22]. The gene *Slc2a11b* plays an important role in the growth and differentiation of yellow and white pigment cells, which is consistent with previous studies on medaka [23]. GTP cyclohydrolase (*Gch*) is an important cofactor of nitric oxide synthase, a rate-limiting enzyme in pteridine synthesis, and catalyzes the de novo synthesis of tetrahydrobioptrexate (H4 bioptera) by *GTP5*. The expression of *gch2* is the initial step of melanocyte and yellow pigment cells differentiation [1], and may play an important role in goldfish pigment aggregation. Furthermore, *mlana* and *loc113079820* are essential pigment agglutination genes in the process of melanosome production and maintenance [24]. Among these 19 genes, the expression level of five genes is the highest in the brown group, and two genes are related to the synthesis and regulation of melanin. The gene *loc113068772* involves myosin, which is related to melanosome transport and melanin aggregation [25,26]; *tyrp1a* involves in dihydroxyindole carboxylic acid oxidase and participates in the production of melanin [8]. The expression levels of the remaining two genes were the highest in the white group, one was not annotated, and the other was related to *Sec62*. Consistent with the above functions, transcriptome sequencing showed that tyrp1 and mlana were significantly upregulated in the black group, while slc2a11b and gch2 were significantly down regulated. In a word, eight genes related to the production of pigment could be selected from common differentially expressed genes as candidate genes, which are worthy of future analysis.

## 5. Conclusions

In conclusion, the current study paints thorough transcriptome profiles of different skin color groups (white, yellow and brown) in celestial goldfish, and eight candidate genes were selected as important functional genes involved in the color variation, including *tyrp1a*, *slc2a11b*, *mlana*, *gch2*, *loc113060382*, *loc113079820*, *loc113068772* and *loc113059134*. Many differentially expressed genes are also enriched in important biological pathways related to pigment production, including *MAPK* signaling pathway, *Wnt* signaling pathway, *tyr*osine metabolism, melanogenesis, etc. Our findings lay the groundwork for future research on the molecular mechanisms behind skin color by identifying possible regulators that may be associated with pigment production in goldfish.

## Figures and Tables

**Figure 1 biology-12-00007-f001:**
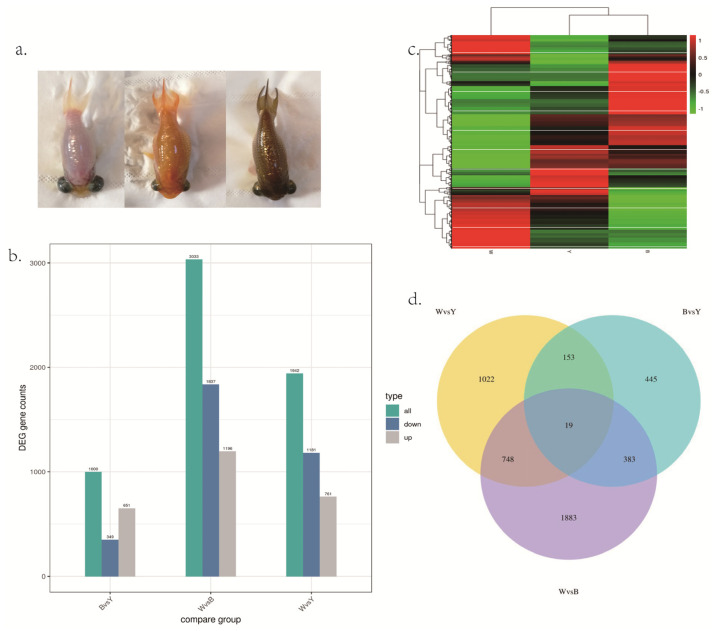
(**a**). The different skin colors in the celestial goldfish. Left: white (W); center: yellow (Y); right: brown (B). (**b**) The bar graph of number of DEGs among three different skin colors. (**c**) Clustering heat map of DEGs in the three skin color groups. (**d**) DEGs number and Venn diagram of the overlap of the different skin color groups. W: white; Y: yellow; B: brown.

**Figure 2 biology-12-00007-f002:**
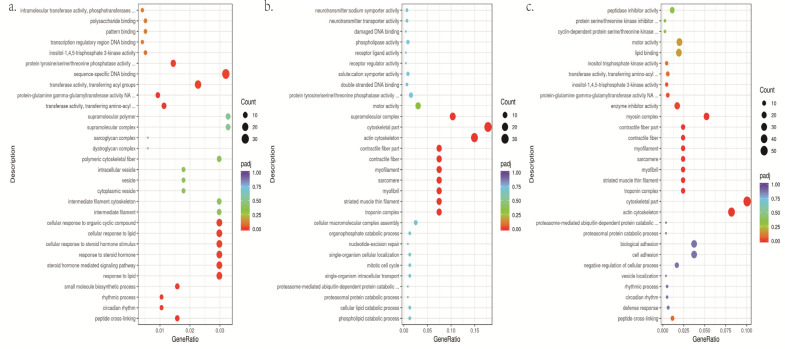
(**a**). The scatter figure of GO enrichment in white vs. yellow. (**b**) The scatter figure of GO enrichment of DEGs in brown vs. yellow. (**c**) The scatter figure of GO enrichment of DEGs in white vs. brown. Count: number of genes in each pathway. GeneRatio: ratio of the number of target genes divided by the total number of genes in each pathway.

**Figure 3 biology-12-00007-f003:**
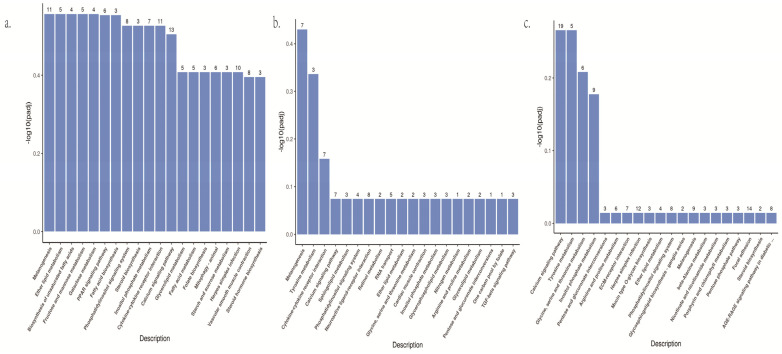
(**a**). The bar graph of KEGG pathway enrichment of DEGs in white vs. yellow. (**b**) The bar graph of KEGG pathway enrichment of DEGs in brown vs. yellow. (**c**) The bar graph of KEGG pathway enrichment of DEGs in white vs. brown.

**Figure 4 biology-12-00007-f004:**
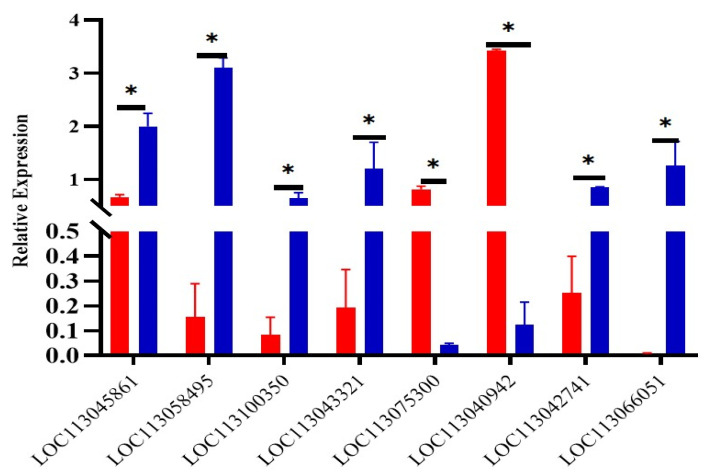
The expression levels of mRNA verified by qPCR. *LOC113045861*-*LOC113043321*: left: yellow group, right: brown group; *LOC113075300*: left: white group, right: brown group; *LOC113040942*-*LOC113066051*: left: yellow group, right: white group. * indicates *p* < 0.05.

**Table 1 biology-12-00007-t001:** A total of 19 common differentially expressed genes among three groups.

Gene ID	FPKM			Description
	White	Yellow	Brown	
113116617	0.053316	14.85341	4.939858	uncharacterized
113059134	0.101642	11.21777	3.472121	perilipin-3-like
novel.91	0.024732	41.3923	11.47289	--
113094292	0	22.96619	9.53373	putative defense protein 3
113054403	0	7.702924	1.333329	diacylglycerol O-acyltransferase 2
113060382	0.021323	4.63168	1.063391	retinol dehydrogenase 7-like
113072589	0	3.77947	0.555086	solute carrier family facilitated glucose transporter member 11-like
113081155	0.065602	12.08961	3.294872	melanoma antigen recognized by T-cells 1-like
novel.919	0.052276	2.696407	9.566884	--
113056131	0.017281	4.126237	0.910635	GTP cyclohydrolase 1-like
113079820	0.023412	8.795228	2.98262	melanoma antigen recognized by T-cells 1-like
113116699	18.29243	526.6035	80.16275	translation initiation factor IF-2-like
113044679	0.872271	0.010594	0.200235	uncharacterized
113107421	0.154702	2.39033	0.707736	RAS and EF-hand domain containing
113059190	1.056176	0.085419	5.333497	transcription elongation factor A protein 3-like
113068772	0.148587	0.020619	1.319104	myosin-6-like
113119606	3.652859	1.230851	0.193719	translocation protein *SEC62*-like
113106535	0.017673	0.195103	1.180688	C6-dihydroxyindole-2-carboxylic acid oxidase-like
novel.58	0.154461	0.976559	4.569909	--

## Data Availability

The datasets provided in this study have been submitted to the online NCBI database (GSE220201).

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
