# Peer review of "Comprehensive Transcriptome Analysis of Different Skin Colors to Evaluate Genes Related to the Production of Pigment in Celestial Goldfish"

_biology, 2022, doi:10.3390/biology12010007_

Round 1
Reviewer 1 Report
Goldfish is a good studying model for phenotypic traits in fish. This manuscript aimed to investigate the candidate genes involved into the formation of pigment in celestial goldfish. The experiment was well designed and provided basic data for understanding goldfish skin color trait.
There are some minor improvement/corrections:
(1) Introduction part: Please cite the latest references (PMID: 35419737) for elucidating the advantages of RNA-seq of skin color traits in goldfish.
(2) Discussion part: This section may require some in-depth mining and give more words for the meaning of the results. For example, no significant difference was revealed between the white and yellow skin, however, some differences were obviously exhibited in Fig 1c, 1d.
(3) There are some mistakes in the manuscript, such as grammar and format.
In paragraph 3, line 6: “obtaine” should be “obtain”.
In Paragraph 3, line 12: “the” should be deleted.
In 3.1, line 1: “The whole-transcriptome” should be “The transcriptome”.
In 3.1, line 11: “inproduction” should be “in production”
(4) Figure: higher resolution figure should be provided and the legend should be listed below the image.
(5)Table: please confirm that the column of "Supplemental Materials" in the text is complete.
(6) Data availability statement: The transcriptomic raw data should be submitted to NCBI and get an available accession
Reviewer 2 Report
Comments and Suggestions for Authors
Li et al. conducted the high-throughput sequencing on the back skin tissues of celestial goldfish with different skin colors, the sequencing work and bioinformatics analysis were done in a routine manner. This manuscript focused on pigment related genes to further understand the expression characteristics of these genes. Some results are interesting and novel. However, some statements of detailed analyses and figure description could be improved to make all manuscript more sense.
Major comments/questions:
Line 71: ‘All three groups of goldfish came from half-sibs’, why not use different color strains?
The fish were 4 months old, provide the weight. And whether the body color is stable at this stage? Why not use the adult fish?
In Fig .1, all text in the picture looks blurry and it is recommended to adjust it to improve clarity, including the horizontal and vertical coordinates and figure legend.
The text of all figures is suggested to be adjusted to improve clarity.
In Fig .4, text and picture overlap, please modify it. And I think there were triplicates of the samples, why not error bar here?
Line 181: ‘8 genes were reported to be related to pigment production and regulation’. References should be supplemented here.
Line 246: ‘there are 19 common differentially expressed genes in three groups, most of which are related to pigment production’. there are 19 common differentially expressed genes related to pigment production, why did you only choose eight? Why were the remaining genes not selected?
Minor comments/questions:
Line 290: It would be preferable to present the actual number of the licence.
Line 118: p should be in italics, Check this throughout the manuscript as this mistake was repeated several times.
Line 311: Species names should be italicized.
Reviewer 3 Report
Line 59: Change “obtaine” to “obtain”
Line 68: What sizes of the goldfish in each group?
Line 71: Authors reported that all three groups of goldfish came from half-sibs. Please explain why half-sibs were used in this study? For half-sibs, which paternal or maternal fish is the same for all three groups of goldfish that were used in this study? What skin colors of paternal and maternal fish in each group of goldfish?
Line 74: Change “80°C” to “-80°C”.
Line 79: Any step to remove DNA contamination from RNA samples before library construction. If yes, please explain.
Line 109: Change “B” to “b”.
Line 136: Please change “Compared with the yellow group” to “When compared between brown and yellow groups”.
Line 138: Please change “Compared with the white group” to “When compared between brown and white groups”. The information of white vs yellow is missing. Please add details about the numbers of genes when compared between white and yellow groups.
Fig. 1b: Please revise all numbers on the bar graph because all numbers are not clear.
Fig. 2a-c and Fig. 3a-c: What is “padj” stand for?
Table 1: Change “a total” to “A total”. What is FPKM and all numbers in each group stand for? Please change “W” to “White”, “Y” to “Yellow”, and “B” to “Brown”.
Fig. 4: Authors reported that the gene expression trends of ten genes obtained by q-PCR was basically consistent with the results of RNA-seq. However, the trends of q-PCR and RNA-seq of LOC113111009 and LOC13073657 genes are not likely to be consistent. Any explanation for this? What are the functions of these genes? In addition, please add the error bar to each bar graph of q-PCR data.
Line 263: What is the function of “Sec62” in fish?
Round 2
Reviewer 2 Report
It can be accepted.
Reviewer 3 Report
The manuscript is suitable to be published.